# VISION FOUNDATION MODELS CAN BE GOOD TOKENIZERS FOR LATENT DIFFUSION MODELS

## ABSTRACT

While latent diffusion models (LDMs) have demonstrated remarkable success in visual generation, the visual tokenizers has proven crucial for effective LDM training. While recent advances have explored incorporating Vision Foundation Model (VFM) representations into visual tokenizers through distillation, our experiment suggests representation degradation happened to these methods. In this paper, we propose a more straight-forward approach to directly leverage frozen VFM encoders within the VAE architecture, proposing Vision Foundation Model Variational Autoencoder (**VFM-VAE**). To address the tension between semantic richness and reconstruction fidelity, we introduce Multi-Scale Latent Fusion and Progressive Resolution Reconstruction blocks in VFM-VAE decoder, enabling high-quality image reconstruction from semantically-rich but spatially-coarse VFM representations. Furthermore, we present a comprehensive analysis of representation dynamics during diffusion training, introducing SE-CKNNA metric and exploring the representation relationship between visual tokenizer and LDMs. Our visual tokenizer design and analysis translates into superior generative performance: our diffusion model reaches a **generation FID of 2.2 without CFG at merely 80 epochs** which is a 10× speedup over prior visual tokenizers. With extra alignment within LDMs, VFM-VAE further attains **1.62 FID at 640 epochs**, establishing direct VFM integration as a superior paradigm for LDMs.

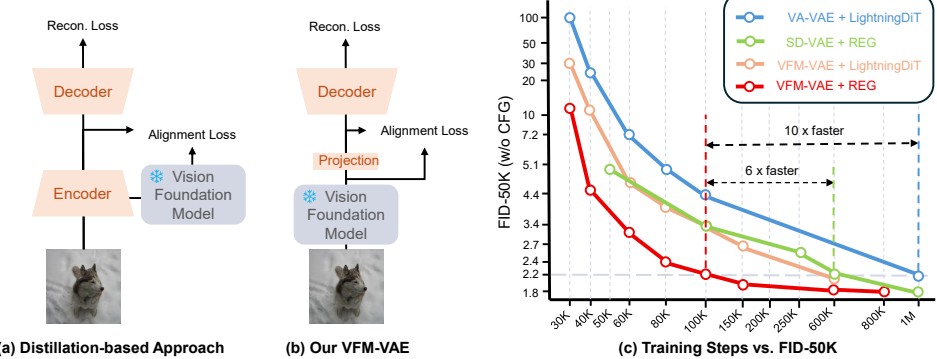

Figure 1: **Comparison of VFM-VAE and Previous Visual Tokenizers for LDM. (a)** Distillation-based approach: VAE variants Yao et al. (2025); Leng et al. (2025) distill advanced representation from VFM. **(b)** Our VFM-VAE: directly leverage frozen VFM as a part of VAE. **(c)** Combing our visual tokenizer with LDMs variants leads faster converge and advanced performance.

## 1 INTRODUCTION

Latent Diffusion Models (LDMs) (Rombach et al., 2022) have emerged as the dominant paradigm in visual synthesis, achieving state-of-the-art performance through an elegant two-stage framework: first train a visual tokenizer (typically a Variational Autoencoder) (Kingma & Welling, 2013) that encodes high-dimensional images into a compact latent space, then learn the diffusion process within

this learned representation space. This approach has proven to be remarkably effective in training high-quality scalable generative models while significantly reducing computational requirements.

The quality of latent representations produced by the visual tokenizer is crucial to the success of the diffusion process. Recently, numerous works (Yao et al., 2025; Leng et al., 2025; Li et al., 2024b; Yang et al., 2025) have explored incorporating Vision Foundation Model (VFM) representations into visual tokenizers, motivated by the impressive progress in self-supervised (Oquab et al., 2023; Siméoni et al., 2025) and weakly-supervised (Radford et al., 2021; Zhai et al., 2023; Tschannen et al., 2025) representation learning. For instance, VA-VAE (Yao et al., 2025) aligns VAE latents with VFM features through a carefully designed similarity loss, while REPA-E (Leng et al., 2025) jointly trains VAE and diffusion models to achieve VFM representation alignment in the diffusion model.

Despite these promising developments, we identify a fundamental limitation in the existing approaches: *alignment-based distillation inevitably introduces representation degradation* compared to the original VFM. Our empirical analysis reveals that aligned representations exhibit unexpected brittleness to semantic preservation transformations(Figure 2), indicating loss of critical information during the distillation process.

This observation motivates our key insight: rather than training a VAE to mimic VFM representations through distillation, we should directly utilize frozen VFM encoders within the VAE framework. While conceptually straightforward, this approach faces a significant challenge: VFMs are optimized for semantic understanding rather than pixel-level reconstruction, creating a fundamental tension between semantic richness and reconstruction fidelity when paired with standard decoders.

Inspired by the success of VFMs in dense prediction tasks (Bolya et al., 2025), we hypothesize that frozen VFM encoders can enable high-fidelity image reconstruction with appropriate architectural adaptations. To this end, we systematically enhance the standard VAE decoder with two key innovations: **Multi-Scale Latent Fusion** and **Progressive Resolution Reconstruction Blocks**. The Multi-Scale Latent Fusion mechanism effectively leverages the hierarchical information inherent in VFM features across different semantic levels, while the Progressive Resolution Reconstruction Blocks enable stable training through advanced synthesis architectures and multi-resolution supervision. This design specifically addresses the challenge of reconstructing pixel-accurate images from semantically-rich but spatially-coarse VFM representations. By combining these architectural innovations with a comprehensive training objective that balances semantic preservation and reconstruction fidelity, we present **Vision Foundation Model Variational Autoencoder (VFM-VAE)**, the first framework to successfully achieve high-quality image reconstruction directly from frozen VFM encoders, eliminating the need for distillation-based alignment.

Building upon VFM-VAE, we conduct a comprehensive study of representation dynamics during diffusion training. We extend the commonly-used representation distance metric CKNNA Huh et al., 2024 to SE-CKNNA (Semantic-Equivariant CKNNA), a more precise metric that better captures semantic equivalence between representations. Our analysis uncovers a previously unidentified *representation degeneration* phenomenon: even when representations align well as denoising targets, their semantic distance from VFM features increases during training. By incorporating representation-aware supervision, we achieve remarkable efficiency gains: our model reaches 2.03 gFID within 64 training epochs, representing a 10× speedup over previous state-of-the-art methods.

Our contributions are threefold:

- We propose VFM-VAE, the first framework to directly leverage VFM encoders for latent diffusion, eliminating distillation-induced degradation while maintaining high reconstruction quality through specialized decoder architectures.

- We present a systematic analysis of representation dynamics in diffusion training, introducing the SE-CKNNA metric and identifying the representation degeneration phenomenon with corresponding solutions.

- We demonstrate state-of-the-art results on ImageNet $256 \times 256$: VFM-VAE with alignment in the generative model, could achieve 1.62 FID without CFG.

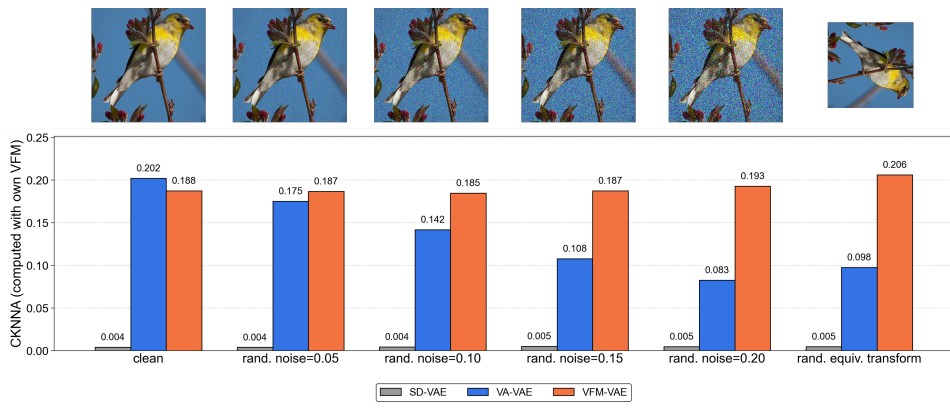

Figure 2: **Brittleness of aligned representations under semantic-preserving transformations.** VFM-VAE shows markedly stronger alignment with VFM features than VA-VAE under semantically invariant perturbations.

## 2 PRELIMINARIES

**Latent Diffusion Models.** We briefly review latent diffusion models through the perspective of *stochastic interpolants* (Albergo et al., 2023; Ma et al., 2024).

Latent diffusion models operate in a two-stage framework. First, a visual tokenizer (typically a VAE) compresses high-dimensional pixel space into a lower-dimensional latent space. The VAE consists of an encoder $\mathcal{E}$ and decoder $\mathcal{D}$, trained jointly to minimize: $\mathcal{L}_{\text{VAE}}(\mathcal{E}, \mathcal{D}) = \mathcal{L}_{\text{KL}} + \mathcal{L}_{\text{MSE}} + \mathcal{L}_{\text{LPIPS}} + \mathcal{L}_{\text{GAN}}$, where the loss combines KL divergence, mean squared error, perceptual loss, and adversarial loss terms.

Given the trained encoder $\mathcal{E}$, we map data $\mathbf{x} \sim p(\mathbf{x})$ to latent representations $\mathbf{z} = \mathcal{E}(\mathbf{x})$. The diffusion process is then defined in this latent space through a time-dependent interpolation between encoded data $\mathbf{z}_* = \mathcal{E}(\mathbf{x}_*)$ and Gaussian noise $\boldsymbol{\epsilon} \sim \mathcal{N}(\mathbf{0}, \mathbf{I})$: $\mathbf{z}_t = \alpha_t \mathbf{z}_* + \sigma_t \boldsymbol{\epsilon}$, where $\alpha_0 = 1$, $\sigma_0 = 0$, $\alpha_T = 0$, $\sigma_T = 1$. Here, $\alpha_t$ and $\sigma_t$ are monotonically decreasing and increasing functions of $t \in [0, T]$, respectively. For linear interpolants, we use $\alpha_t = 1 - t$ and $\sigma_t = t$ with $T = 1$.

The latent diffusion model learns a velocity field $\mathbf{v}_\theta(\mathbf{z}_t, t)$ by minimizing the following objective: $\mathcal{L}_v(\theta) = \mathbb{E}_{\mathbf{x}_*, \boldsymbol{\epsilon}, t} \left[ \|\mathbf{v}_\theta(\mathbf{z}_t, t) - (\dot{\alpha}_t \mathbf{z}_* + \dot{\sigma}_t \boldsymbol{\epsilon})\|^2 \right]$, where $\mathbf{z}_* = \mathcal{E}(\mathbf{x}_*)$, $\mathbf{z}_t$ is constructed according to the interpolation formula, and $\dot{\alpha}_t, \dot{\sigma}_t$ denote time derivatives.

At inference, we generate samples by solving the probability flow ODE: $\frac{d\mathbf{z}_t}{dt} = \mathbf{v}_\theta(\mathbf{z}_t, t)$, $\mathbf{z}_0 \sim \mathcal{N}(\mathbf{0}, \mathbf{I})$, integrating from $t = 0$ to $t = T$. The final image is obtained by decoding: $\hat{\mathbf{x}} = \mathcal{D}(\mathbf{z}_T)$. As evident from these formulations, the visual tokenizer plays a crucial role in both the diffusion training process and the final image reconstruction during inference.

## 3 METHOD: VISION FOUNDATION MODEL VAE

### 3.1 OVERVIEW

To address the fundamental limitations of alignment-based distillation approaches, we propose Vision Foundation Model Variational Autoencoder (VFM-VAE), which directly leverages frozen pretrained VFM encoders while learning specialized decoders for high-fidelity reconstruction. As illustrated in Figure 3, our framework comprises three key components: (1) a VFM-VAE encoder that serves as a lightweight wrapper around the pre-trained VFM, (2) a carefully designed VFM-VAE decoder that captures dense hierarchical information from the encoder to reconstruct high-quality images, and (3) a comprehensive training objective that balances semantic preservation with reconstruction fidelity. We describe each component in detail below.

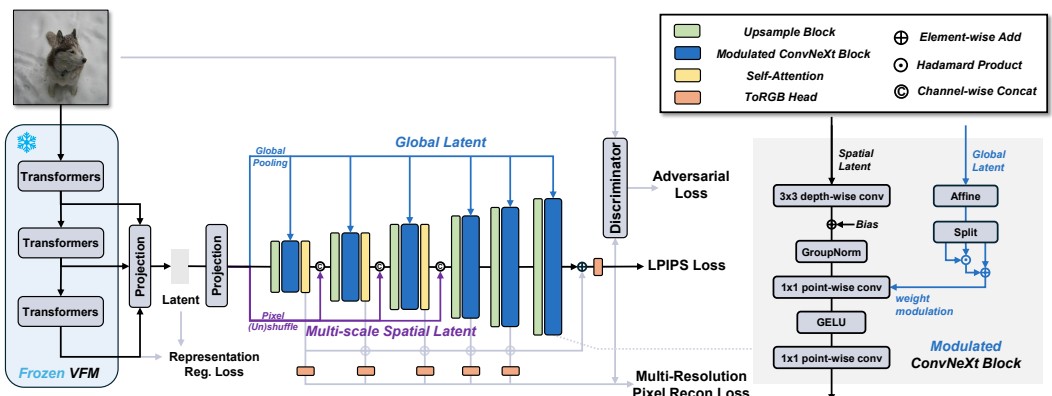

Figure 3: **Overview of VFM-VAE architecture design.**

## 3.2 VFM-VAE ENCODER ARCHITECTURE

Unlike previous approaches that train VAE encoders from scratch to align with VFM representations, we directly leverage a pre-trained VFM as our encoder $\Phi$, keeping it frozen throughout training to preserve its rich semantic representation extraction ability. Following insights from literatures where VFM is used for dense prediction tasks (Liu et al., 2023; Bolya et al., 2025), we recognize that optimal features for reconstruction may not reside solely in the final layer. Therefore, we extract multi-scale features from different depths of the VFM hierarchy.

Given an input image $\mathbf{x} \in \mathbb{R}^{H \times W \times 3}$, the VFM encoder extracts features at multiple layers:

$$\{\mathbf{f}_{\text{shallow}}, \mathbf{f}_{\text{middle}}, \mathbf{f}_{\text{final}}\} = \Phi(\mathbf{x}) \tag{1}$$

where these features correspond to the first, the middle, and final layers of the VFM, capturing different levels of semantic understanding and spatial detail.

Similar to usual VAE for latent diffusion, we need to further reduce the latent's dimension to facilitate the learning of diffusion model. Hence we then concatenate these multi-scale features along the channel dimension and apply a learnable light-weight projection module:

$$\mathbf{z} = \mathcal{C}(\text{Concat}[\mathbf{f}_{\text{shallow}}, \mathbf{f}_{\text{middle}}, \mathbf{f}_{\text{final}}]) \tag{2}$$

where $\mathcal{C}$ is a lightweight projection network that reduces dimensionality. We also add a representation reconstruction loss to ensure the latent preserves essential information of original VFM high-dimension representation, which will be introduced in the following subsection. This encoder design is crucial for maintaining computational efficiency in subsequent diffusion training while retaining the semantic richness of VFM representations.

## 3.3 VFM-VAE DECODER ARCHITECTURE

Different from the commonly-used SD-VAE decoder architecture using single latent input and single image output, we systematically upgrade it with Multi-Scale Latent Fusion and Progressive Resolution Reconstruction Blocks, which is designed to address the challenge of reconstructing high-fidelity images from semantically-rich but detail-lost VFM representations. We sequentially introduce Multi-Scale Latent Fusion for decoder inputs and Progressive Reconstruction Blocks for network details.

**Multi-Scale Latent Fusion.** Given the latent representation $\mathbf{z}$, we first decompose it into global and spatial components to enable scale-specific processing: $\mathbf{z}_g = \text{GlobalPool}(\mathbf{z}) \in \mathbb{R}^c$ and $\mathbf{z}_s^{(i)} = \text{Reshape}_i(\mathbf{z}) \in \mathbb{R}^{c \times h_i \times w_i}$, where $\mathbf{z}_g$ captures global style information, and $\{\mathbf{z}_s^{(i)}\}_{i=1}^N$ represent spatial features be reshaped at different scales with $Reshape_i$ operations including pixels shuffle and unshuffle (Shi et al., 2016). These multi-scale latents will be well utilized in the following progressive reconstruction blocks.

**Progressive Reconstruction Blocks.** To enhance latent representation capture and detail synthesis capability, we redesign the decoder basic blocks architecture to reconstruct images at multiple reso-

lutions. The decoder consists of a series of $N$ blocks $\{\mathcal{B}_i\}_{i=1}^N$, each responsible for upsampling and refining features to a higher resolution:

$$\mathbf{h}^{(i)} = \mathcal{B}_i(\text{Concat}[\mathbf{h}^{(i-1)}, \mathbf{z}_s^{(i)}], \mathbf{z}_g) \quad \text{for } i \leq 3 \tag{3}$$

where $\mathbf{h}^{(0)} = \mathbf{z}_s^{(1)}$ for the first block, and $\mathbf{z}_s^{(i)}$ is only available for the first three blocks. Each block employs modulated convolutions (Karras et al., 2019) to inject global style information:

$$\mathcal{B}_i(\mathbf{h}, \mathbf{z}_g) = \text{ModConv}(\mathbf{h}, \gamma_i(\mathbf{z}_g)) + \mathbf{h} \tag{4}$$

where $\gamma_i$ is a learned affine transformation that converts global features into modulation parameters.

Unlike traditional decoders that produce output only at the final resolution, we attach reconstruction heads at each scale, where a residual connect is added with features from previous stage:

$$\mathbf{x}_i = \text{ToRGB}_i(\mathbf{h}^{(i)} + \mathbf{h}^{(i-1)}) \in \mathbb{R}^{r_i \times r_i \times 3} \tag{5}$$

where $r_i$ denotes the output resolution at block $i$. This multi-resolution output enables progressive supervision during training and ensures that each block learns appropriate level of detail for its scale.

### 3.4 TRAINING OBJECTIVES

Besides reconstructing the original detailed image, though we have frozen the VFM, we have to ensure the reduced dimension latent $\mathbf{z}$ still preserve as much as possible information originally contained. Hence, our training objective combines representation preservation with multi-resolution reconstruction to achieve both semantic consistency and pixel-level fidelity: $\mathcal{L}_{\text{total}} = \lambda_{\text{rep}}\mathcal{L}_{\text{rep}} + \sum_{i=1}^N \lambda_i \mathcal{L}_{\text{recon}}^{(i)} + \lambda_{\text{GAN}}\mathcal{L}_{\text{GAN}} + \lambda_{\text{LPIPS}}\mathcal{L}_{\text{LPIPS}}$.

**Representation Regularization Loss.** To ensure that our projected latents maintain semantic alignment with the original VFM representation, we have to assign $\mathcal{L}_{\text{rep}}$ as a regularizor. In our implementation we directly leverage VF loss (Yao et al., 2025). It calculates cosine similarity and matrix distance between two representations, which is specifically crafted to regularize high-dimensional latent spaces without overly constraining their capacity. Combining the Kullback-Leibler divergence loss used in previous visual tokenizer for latent diffusion, we have $\mathcal{L}_{\text{rep}} = \mathcal{L}_{\text{VF}}(\mathbf{z}, \mathbf{f}_{\text{final}}) + \mathcal{L}_{\text{KL}}(\mathbf{z})$.

**Multi-Resolution Reconstruction Loss.** Each resolution receives direct supervision: $\mathcal{L}_{\text{recon}}^{(i)} = \|\mathbf{f}_{r_i}(\mathbf{x}) - \mathbf{x}_i\|_1$, where the $\mathbf{f}_{r_i}$ denotes the resize function to a specified resolution of block $i$ output image. The multi-scale supervision ensures stable training and prevents mode collapse at early stages.

**Adversarial and Perceptual Losses.** Following recent advances in adversarial training image synthesis (Karras et al., 2019; Sauer et al., 2024), we incorporate adversarial training with a DINOv2-base backbone with discrimination head and LPIPS loss (Zhang et al., 2018) for the final full-resolution output to enhance perceptual quality.

The proposed VFM-VAE architecture effectively balances semantic representation quality with reconstruction fidelity, providing an ideal foundation for subsequent diffusion model training.

## 4 EXPERIMENTS

In this section, we first evaluate VFM-VAE as a visual tokenizer, then comprehensively analyze how the representations produced by different tokenizers affect diffusion model training, both with and without explicit alignment losses. Finally, we report generation performance when combining VFM-VAE with various diffusion models. The content is organized as follows:

- **Reconstruction & Latent Representation:** We assess how our architectural choices contribute to high-fidelity image reconstruction and examine the impact of our visual tokenizer on latent representations (Table 2, 3, 4 Figure 6).
- **Representation Diagnose in Diffusion Models:** We investigate how the choice of visual tokenizer influences representation learning within LDMs (Figure 2, 4).
- **Generation Performance:** We evaluate whether integrating VFM-VAE with generative models accelerates convergence and improves sample quality (Table 1, Figure 5).

Table 1: **System-level generative performance on ImageNet 256×256.**

| Tokenizer | Method | Training Epochs | #params | rFID↓ | Generation w/o CFG gFID↓ | IS↑ | Generation w/ CFG gFID↓ | sFID↓ | IS↑ | Prec.↑ | Rec.↑ |
|---|---|---|---|---|---|---|---|---|---|---|---|
| **AutoRegressive (AR)** | | | | | | | | | | | |
| MaskGiT | MaskGiT (Chang et al., 2022) | 555 | 227M | 2.28 | 6.18 | 182.1 | - | - | - | - | - |
| VQGAN (Yu et al., 2021) | LlamaGen (Sun et al., 2024) | 300 | 3.1B | 0.59 | 9.38 | 112.9 | 2.18 | 5.97 | 263.3 | 0.81 | 0.58 |
| VQVAE (Yao et al., 2025) | VAR (Tian et al., 2024) | 350 | 2.0B | - | - | - | 1.80 | - | **365.4** | **0.83** | 0.57 |
| LFQ tokenizers | MagViT-v2 (Yu et al., 2023) | 1080 | 307M | 1.50 | 3.65 | 200.5 | 1.78 | - | 319.4 | - | - |
| LDM (Rombach et al., 2022) | MAR (Li et al., 2024a) | 800 | 945M | 0.53 | 2.35 | 227.8 | 1.55 | - | 303.7 | 0.81 | 0.62 |
| **Latent Diffusion Models (LDM)** | | | | | | | | | | | |
| SD-VAE | MaskDiT (Zheng et al., 2023) | 1600 | 675M | 0.58 | 5.69 | 177.9 | 2.28 | 5.67 | 276.6 | 0.80 | 0.61 |
| | DiT (Peebles & Xie, 2023) | 1400 | 675M | | 9.62 | 121.5 | 2.27 | 4.60 | 278.2 | **0.83** | 0.57 |
| | SiT (Ma et al., 2024) | 1400 | 675M | | 8.61 | 131.7 | 2.06 | 4.50 | 270.3 | 0.82 | 0.59 |
| | FasterDiT Yao et al. (2024) | 400 | 675M | | 7.91 | 131.3 | 2.03 | 4.63 | 264.0 | 0.81 | 0.60 |
| | MDT (Gao et al., 2023a) | 1300 | 675M | | 6.23 | 143.0 | 1.79 | 4.57 | 283.0 | 0.81 | 0.61 |
| | MDTv2 (Gao et al., 2023b) | 1080 | 675M | | - | - | 1.58 | 4.52 | 314.7 | 0.79 | 0.65 |
| **Representation Alignment Methods** | | | | | | | | | | | |
| E2E-VAE (Leng et al., 2025) | REPA (Yu et al., 2024) | 80 | 675M | **0.28** | 3.46 | 159.8 | 1.67 | 4.12 | 266.3 | 0.80 | 0.63 |
| | | 800 | 675M | | 1.83 | 217.3 | **1.26** | **4.11** | 314.9 | 0.79 | **0.66** |
| VA-VAE (Yao et al., 2025) | LightningDiT (Yao et al., 2025) | 64 | 675M | 0.30 | 5.14 | 130.2 | 2.11 | 4.16 | 252.3 | 0.81 | 0.58 |
| | | 80 | 675M | | 4.29 | | - | - | - | - | - |
| | | 800 | 675M | | 2.17 | 205.6 | 1.35 | 4.15 | 295.3 | 0.79 | 0.65 |
| SD-VAE | REPA | 80 | 675M | 0.58 | 7.90 | 122.6 | - | - | - | - | - |
| | | 800 | 675M | | 5.90 | 157.8 | 1.42 | 4.70 | 305.7 | 0.80 | 0.65 |
| | REG (Wu et al., 2025) | 80 | 675M | | 3.40 | 184.1 | 1.86 | 4.49 | 321.4 | 0.76 | 0.63 |
| | | 480 | 675M | | 2.20 | 219.1 | 1.40 | 4.24 | 296.9 | 0.77 | **0.66** |
| VFM-VAE | LightningDiT | 64 | 675M | 0.52 | 3.80 | 152.8 | 2.16 | 4.26 | 232.8 | 0.82 | 0.58 |
| | | 80 | 675M | | 3.41 | 160.4 | - | - | - | - | - |
| | | 560 | 675M | | 2.06 | 205.8 | 1.57 | 4.56 | 254.4 | 0.80 | 0.64 |
| | REG | 64 | 685M | | 2.42 | 215.2 | 2.03 | 5.23 | 261.7 | **0.83** | 0.58 |
| | | 80 | 685M | | 2.22 | 218.8 | - | - | - | - | - |
| | | 480 | 685M | | 1.67 | 238.3 | 1.34 | 4.59 | 302.7 | 0.78 | 0.65 |
| | | 640 | 685M | | **1.62** | **241.6** | 1.31 | 4.63 | 300.2 | 0.78 | **0.66** |

Table 2: **Reconstruction, generation, and representation metrics.** The generation results are obtained by training LightningDit-XL (Yao et al., 2025) 64 epochs on ImageNet. Details of SE-CKNNA metric is described in Sec. 4.3.

| Method | Training Images | Reconstruction rFID↓ | rIS↑ | Generation gFID↓ | gIS↑ | Representation CKNNA | SE-CKNNA |
|---|---|---|---|---|---|---|---|
| SD-VAE | 60M | 0.58 | 207.6 | 17.20 | - | 0.003 | 0.005 |
| VA-VAE | 160M | **0.30** | **213.6** | 5.14 | 130.2 | 0.109 | 0.135 |
| VFM-VAE | 44M | 0.52 | 208.0 | **3.80** | **152.8** | **0.111** | **0.191** |

## 4.1 SETUP

**Baselines.** For visual tokenizers, besides commonly-used SD-VAE (Rombach et al., 2022), we introduce mainstream alignment-based methods including VA-VAE (Yao et al., 2025) and REPA-E (Leng et al., 2025). For generative models, our comparison involves the works in the REPA (Yu et al., 2024) and the latest REG (Wu et al., 2025), which align intermediate layers of diffusion transformers with VFM features to accelerate feature learning and convergence.

**Implementation details.** Both tokenizer training and generative model training are conducted on the ImageNet dataset (Russakovsky et al., 2015). In all experiments, the image resolution is fixed at 256, and VFM-VAE consistently adopts the same f16d32 setting as VA-VAE (Yao et al., 2025). For generative models paired with VFM-VAE, we consider two settings: (1) LightningDiT-XL, with configurations and parameters identical to those of the VA-VAE baseline; and (2) the REG model using SiT-XL (Ma et al., 2024) with extra alignment loss. Additional experimental details, including hyperparameter settings and computing resources, are provided in Appendix C.

**Evaluation.** For reconstruction performance, we report Fréchet Inception Distance (FID; Heusel et al., 2017) and Inception Score (IS; Salimans et al., 2016) on the ImageNet 50K validation set. For generative performance, we follow the ADM (Dhariwal & Nichol, 2021) setup and report generation FID (gFID), structural FID (sFID; Nash et al., 2021), IS, as well as precision (Prec.) and recall (Rec.) (Kynkäänniemi et al., 2019). The computation of CKNNA follows the protocol described in Huh et al. (2024).

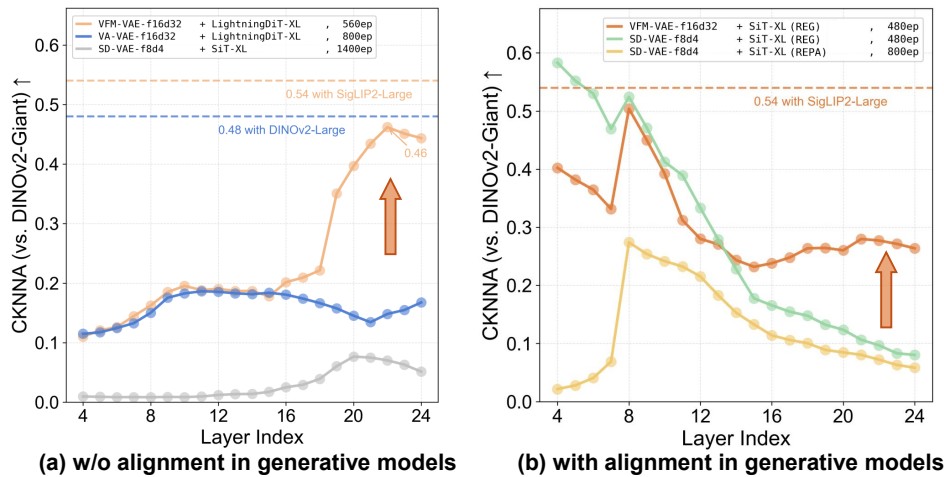

Figure 4: **CKNNA comparison across layers of the generative models.**

## 4.2 RECONSTRUCTION AND LATENT REPRESENTATION

**VFM-VAE achieves semantic-rich representation and impressive reconstruction fidelity.** Reconstruction ability sets the upper bound for the quality of generations, making strong reconstruction metrics the primary requirement for any tokenizer used in LDMs. For qualitative results, we provide a visual comparison between VFM-VAE and mainstream VAE methods in Appendix D.1. VFM-VAE shows no compromise in object structure or natural textures. This observation is also reflected in quantitative metrics (Table 2): with its dual-branch design for semantic and spatial control, VFM-VAE **surpasses SD-VAE** in fidelity-oriented measures such as FID and IS, thus satisfying the prerequisites of an effective tokenizer for LDMs.

Beyond reconstruction quality, we further evaluate alignment by computing the CKNNA metric between VAE latents and the DINOv2-Giant (Oquab et al., 2023), as a unified measure of VFM feature alignment. Remarkably, with only about **25% of the training images** used by VA-VAE, VFM-VAE not only achieves considerable reconstruction performance but also delivers comparable alignment. These results highlight that the gains of VFM-VAE come both from starting with VFM features and from our carefully designed framework, which together balance alignment and reconstruction. Detailed results from ablation studies can be found in Section 4.5.

## 4.3 REPRESENTATION DIAGNOSE IN DIFFUSION MODELS

**CKNNA might not be sufficient to evaluate latent representation of visual tokenizer.** Though CKNNA (Huh et al., 2024) has been studied to evaluate the representation within diffusion models, how well it reflects the effectiveness of visual tokenizers remains underexplored. From observation in Table 2, both VA-VAE and VFM-VAE achieve similar CKNNA scores, yet when integrated into generative models, VFM-VAE outperforms VA-VAE by a significant margin. This discrepancy suggests that CKNNA alone may not fully capture the qualities that make a tokenizer effective for generative modeling.

Inspired by the observation that the diffusion model often encounters suboptimal images that deviate from the clean image distribution during training, we extend CKNNA to **SE-CKNNA** (Semantic-Equivariant CKNNA): we apply a series of semantic-preserving transformation on the evaluated images, then feed the perturbed images into the VFM and VAE to extract features and compute CKNNA. These transformations include noise perturbations of increasing strength and equivariant perturbations (scale-interpolation strengths $\{0.25, 0.5, 0.75, 1.0\}$; discrete rotations $\{0°, 90°, 180°, 270°\}$). The results shown in Table 2 indicate that VFM-VAE achieves a significantly higher SE-CKNNA score compared to VA-VAE, suggesting that VFM-VAE maintains better alignment with VFM features even under these transformations, yielding more robust representations for generative modeling.

**SE-CKNNA of latent representation reflects the representation quality in LDMs.** In Figure 4a, we compare layer-wise CKNNA within generative models trained without explicit alignment. We find that VFM-VAE consistently achieves higher layer-wise CKNNA and superior generative performance. Moreover, while the VA-VAE baseline shows consistently weaker layer-wise CKNNA compared to its alignment upper bound (computed between VFM-DINOv2-Large and VFM-DINOv2-Giant), VFM-VAE combined with LightningDiT reaches a peak CKNNA of 0.46, corresponding to 85% of the upper bound, indicating that it inherits VFM properties more effectively. For direct generation metric, VFM-VAE + LightningDiT achieves an FID of 3.80 at 64 epochs, significantly outperforming the VA-VAE baseline's FID of 5.14 at the same training stage. *These results validate our inference and demonstrate the effectiveness of SE-CKNNA as a measure of how tokenizer alignment translates into gains for generative modeling.*

**Joint tokenizer and LDM alignment yields consistently high-quality representations.** Building on the impressive results achieved without explicit LDM alignment loss, we further investigate the potential of combining VFM-VAE with alignment methods in the generative model. As shown in Figure 4, when we combine VFM-VAE with LightningDiT, alignment in shallow layers remains relatively weak compared to models with explicit alignment, with substantial improvements only emerging around layer 16. Ideally, the generative model should maintain features highly aligned with VFM throughout its depth, preserving strong semantic information while filtering out unnecessary details. This observation motivates a natural question: *can we explicitly enforce VFM alignment in the generative model's shallow layers to ensure consistency across all depths?*

To address this, we incorporate REG (Wu et al., 2025), a state-of-the-art alignment method for generative models. REG extends REPA by not only aligning shallow layers with VFM patch features from clean images but also concatenating the VFM class token with generative tokens, thereby introducing global semantic guidance. As illustrated in Figure 4, the combination of VAE-side alignment via VFM-VAE with shallow-layer alignment in the generative model yields consistently high layer-wise CKNNA across all depths. The average CKNNA across layers significantly exceeds that achieved by applying REG or REPA independently.

## 4.4 GENERATION PERFORMANCE

This alignment advantage also directly translates into improved generative performance (Table 1): under the without CFG setting, VFM-VAE + REG achieves an FID of 2.42 in only 64 epochs, surpassing REG trained for 200 epochs. At 480 epochs, the result further improves to 1.67, outperforming the final outcomes of all compared methods. **Ultimately, after 640 epochs, our model achieves FIDs of 1.62 (without CFG) and 1.31 (with CFG), demonstrating faster convergence and superior performance.**

In summary, through the design of VFM-VAE, the generative model acquires stronger representational learning capacity even when the tokenizer is trained on fewer images, significantly outperforming the VA-VAE baseline. Furthermore, by incorporating alignment at the generative model side, our model achieves state-of-the-art results. These findings highlight the feasibility and effectiveness of maintaining VFM alignment simultaneously in both the latent space of LDM and latent space of the generative model.

In addition, Figure 5 presents a qualitative comparison of generation results across training stages using a fixed random seed and fixed noise. Systems equipped with VFM-VAE consistently produce results that are noticeably superior to the VA-VAE baseline. More strikingly, the model with VFM-VAE + REG, which maintains strong alignment across all layers, is already capable of generating detail-rich and realistic images at only 100k steps (about 80 epochs) under the without CFG setting. This further reinforces our conclusion: maintaining strong and consistent alignment with VFM features directly translates into superior generative performance.

## 4.5 ABLATION STUDIES - ALL COMPOMENTS ARE NECCESSARY.

The overall architecture comprises a frozen VFM, a lightweight encoder, and a decoder. Throughout the design, we employ a dual-branch structure for semantic and spatial control, while continuously adding feature-regularization losses during training to preserve alignment with VFM features. Build-

ing on this foundation, we progressively extend a minimal baseline with additional components and evaluate their effects on reconstruction performance:

- **SD-VAE-Style Baseline.** We begin with a simple VAE design where the VFM is SigLIP2-Large-Patch16-256. The encoder and decoder each contain two convolutional layers for pre- and post-sampling encoding/decoding. The loss functions consist only of L1, LPIPS, and adversarial terms.
- **Multi-scale Latent Fusion.** Building on this baseline, we introduce Multi-scale Latent Fusion as described in Sec. 3.3. We also upgrade the upsampling modules and incorporate multi-scale reconstruction losses to stabilize training and accelerate early convergence.
- **Modern Blocks.** We substitute modulated convolution blocks with modern ConvNeXt-based variants, and insert self-attention at low-resolution stages to better decode semantic features.
- **Encoder Mofications.** We aggregate features from shallow (first layer), intermediate (central layer), and final VFM layer, thereby leveraging extracted information across multiple levels.

As shown in Table 3, we conduct lightweight alignment training on 5M images for a quick evaluation. The minimal baseline produces an FID of 19.79, rendering it nearly unusable. Adding spatial control reduces FID by about 27%. With enhanced encoder/decoder capacity and modernized modules, rFID drops to 1.08, essentially meeting the standard for fast convergence and usable reconstruction. Incorporating additional tricks further reduces rFID below 1, achieving even higher reconstruction fidelity.

Table 3: **VFM-VAE module ablation study**.

| Setting | #params | rFID↓ | IS↑ |
|---|---|---|---|
| SD-VAE-style Baseline | 43.0M | 19.69 | 74.9 |
| + Multi-scale Latent Fusion | 88.0M | 14.35 | 93.6 |
| + Modern Blocks | 132.3M | 1.08 | 194.6 |
| + Encoder Modifications | 140.6M | **0.71** | **206.8** |

In summary, this ablation study demonstrates that the carefully designed VFM-VAE significantly improves reconstruction quality while maintaining VFM feature alignment, with each module contributing an indispensable role. In addition to ablations on architectural components, we also validate the feasibility of integrating VFM-VAE with different VFMs (see Appendix B.2), further highlighting the generality of the VFM-VAE design.

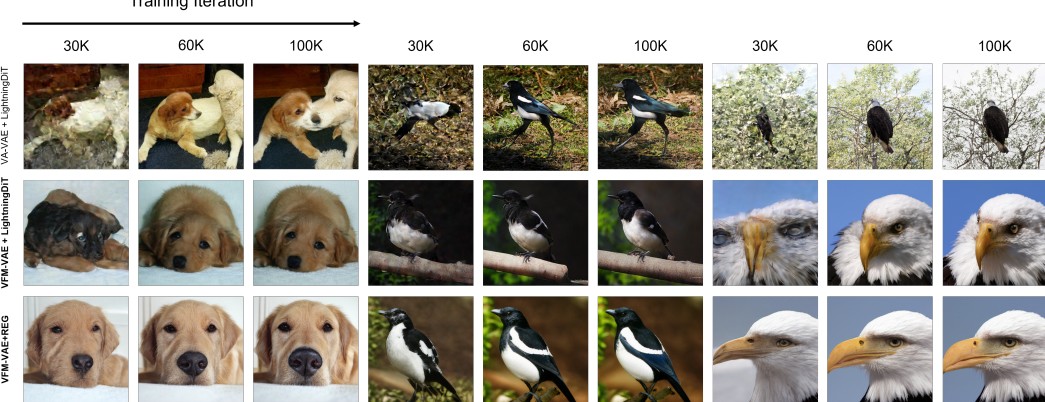

Figure 5: **Stage-wise visualization of generative model training results.** Our approach shows impressive performance to greatly accelerate the learning of image generation.

## 5 CONCLUSION

In this paper, we presented VFM-VAE, a novel framework that directly leverages frozen Vision Foundation Model encoders for latent diffusion models, eliminating the representation degradation inherent in distillation-based alignment approaches with careful architectural design. The effectiveness of our approach is evident in both quantitative and qualitative results. By establishing direct VFM integration as a superior paradigm for visual tokenization, this work opens new avenues for leveraging pre-trained vision models in generative tasks.

ETHICS STATEMENT

This study relies entirely on publicly available and widely used benchmark datasets (e.g., ImageNet) as well as open-source implementations. No personally identifiable information, sensitive content, or proprietary data are involved at any stage of the research. The proposed methods are designed solely to advance academic research in generative modeling and are not intended for deployment in sensitive, security-critical, or high-risk application domains. Apart from the common ethical considerations generally associated with generative model research, such as potential misuse or biased data distributions, we are not aware of any additional direct ethical concerns.

REPRODUCIBILITY STATEMENT

We provide hyperparameter details in Section 4.1 and Appendix C. We will release the implementation and pretrained model checkpoints in the future to ensure the reproducibility of our results.

LLM USAGE

Large language models (ChatGPT) were used only for language refinement of the paper draft. They did not contribute to research ideation, experimental design, or analysis. All technical content, methods, and results were created and verified by the authors.

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

# A    ARCHITECTURAL DETAILS OF VFM-VAE

## A.1    ATTENTION PROJECTION (FROM UNITOK)

**Why we adopt it.** We adopt UniTok (Ma et al., 2025)'s *Attention Projection* because it offers a simple, stable, and compute-efficient way to compress channels before sampling and to re-expand them during decoding, helping maintain a well-behaved representation distribution aligned with VFM features.

**Encoder-side (compression).** We extract shallow, middle, and final features from the frozen VFM. If the spatial size (and thus tokenization) does not match the target latent configuration, we first *shuffle* to reconcile the spatial mismatch, then *concatenate* the tokens. **Critically, the concatenated tokens are passed *once* through the Attention Projection** to project into the latent space, where we compute distribution statistics (e.g., mean/variance) and sample latents.

**Decoder-side (expansion).** During decoding, we reuse the same *Attention Projection* to expand channels and dispatch the outputs to the *semantic* and *spatial* branches of the decoder, enabling the model to recombine abundant low-level cues for subsequent decoding.

## A.2    UPSAMPLING MODULE

This module is an improved version of the StyleGAN-T (Karras et al., 2019) upsampling unit. While the original implementation relies on C-language kernels, our design adopts a pure PyTorch implementation for better readability and extensibility, while also enhancing functionality. Data processing proceeds as follows: the input feature map is first normalized with GroupNorm to stabilize the feature distribution, followed by a depthwise convolution (local feature extraction) and a pointwise convolution (channel expansion). The result is spatially upsampled using PixelShuffle, and finally smoothed with a fixed Gaussian blur kernel to suppress checkerboard artifacts.

Within the network, the module serves two purposes:

- **Backbone upsampling** – progressively enlarging the resolution of the initial input so that subsequent blocks operate on increasingly finer spatial scales.
- **Output pathway** - upsampling the features from the previous block, performing residual accumulation at the feature level, and passing it into the ToRGB head to generate image.

This design preserves the efficiency of StyleGAN-T's upsampling strategy, while improving maintainability, stabilizing intermediate representations, and yielding higher-quality image outputs.

## A.3    TORGB HEAD

Our ToRGB Head inherits from StyleGAN-T (Karras et al., 2019): it first projects the latent $w$ into channel-wise scaling factors through an Affine layer to modulate convolutional weights on a per-sample basis; then, grouped convolution is applied to realize independent modulated feature mapping, and a bias is added at the output to produce the RGB image, thereby combining style control with efficient image-space projection.

# B    ANALYSIS DETAILS

## B.1    CKNNA EVALUATION DETAILS

Centered Kernel Nearest-Neighbor Alignment (CKNNA) (Huh et al., 2024) measures whether two representations preserve the same *local neighborhoods*: given kernel (similarity) matrices $K, L \in \mathbb{R}^{n \times n}$ from two representations of the same $n$ samples, we keep only pairs that are common $k$-nearest neighbors in both spaces via the mask $A_{ij} = \alpha(i, j; k) = \mathbf{1}\{i \neq j, j \in \text{knn}_k^K(i) \wedge j \in \text{knn}_k^L(i)\}$ and define $K^{(k)} = K \odot A$, $L^{(k)} = L \odot A$ (Hadamard product). With the centering matrix

$$H = I - \frac{1}{n}\mathbf{1}\mathbf{1}^\top,$$

we *double-center* each kernel by subtracting row/column means and adding back the grand mean. This operation removes global offsets and overall scale differences, so that the comparison focuses

purely on the covariance of pairwise relations rather than absolute similarity levels. Intuitively, $H$ ensures that the CKNNA score measures whether two representations agree on the *pattern of neighborhood relations* ("who is closer to whom"), instead of being confounded by uniform shifts in similarity values.

The final CKNNA score is then defined as the CKA-style normalized inner product of the centered, locally masked kernels:

$$\mathrm{CKNNA}_k(K, L) = \frac{\left\langle HK^{(k)}H,\ HL^{(k)}H \right\rangle_F}{\sqrt{\left\| HK^{(k)}H \right\|_F^2 \left\| HL^{(k)}H \right\|_F^2}} \in [0, 1].$$

Larger values indicate stronger agreement on "who is close to whom"; smaller $k$ emphasizes the very nearest neighbors, while larger $k$ yields a smoother, more global view.

In the preprocessing step of CKNNA, we follow the configuration of the original work by first applying outlier filtering and channel-wise normalization, and fixing $k$ to 10. When extracting layer-wise CKNNA for generative models, we adopt the same strategy as (Yu et al., 2024) by selecting only spatial tokens (e.g., REG (Wu et al., 2025) discards the [CLS] token) and computing the alignment after global pooling along the spatial dimension for both the target model and the reference features.

## B.2    NOT ONLY ONE VFM CHOICE

VFM features play a critical role in both reconstruction and generation quality. In Table 4, we evaluate EVA-CLIP-Large (Sun et al., 2023), DINOv2-Large (Oquab et al., 2023), and SigLIP2-Large Tschannen et al. (2025) under strong alignment, focusing on their reconstruction and alignment performance. The results show that DINOv2-Large achieves slightly worse reconstruction than SigLIP2-Large, raising an important concern: *could the performance gap between VFM-VAE (aligned with SigLIP2-Large) and VA-VAE (aligned with DINOv2-Large) in generation be attributed merely to differences in the underlying VFMs?* We address this question in Appendix B.3. Notably, although EVA-CLIP-Large surpasses SigLIP2-Large in both reconstruction and alignment, SigLIP2 adopts a more balanced training objective and consistently outperforms EVA-CLIP-Large on downstream tasks. Considering training cost and feature generalization, we ultimately choose SigLIP2 for long-term training. Nevertheless, the strong reconstruction and alignment performance of both EVA-CLIP and DINOv2 indicates that VFM-VAE is not restricted to a single VFM but is broadly compatible with current mainstream VFMs.

Table 4: **Comparison across VFMs.** CKNNA is computed with own VFM.

| VFM | rFID↓ | IS↑ | CKNNA↑ |
|---|---|---|---|
| EVA-CLIP-Large | **2.25** | 172.0 | 0.270 |
| DINOv2-Large | 3.77 | **178.8** | 0.335 |
| SigLIP2-Large | 2.70 | 158.4 | 0.220 |

## B.3    VERIFYING IMPROVEMENTS ARE NOT DUE TO STRONGER VFMS

VFM-VAE consistently demonstrates faster convergence and higher generation quality than the VA-VAE baseline across all generative models. A noteworthy detail, however, lies in the choice of VFMs used for alignment: VFM-VAE is anchored on SigLIP-Large, which is trained primarily with contrastive learning and is capable of both vision–language alignment and dense visual feature learning; in contrast, VA-VAE is aligned with DINOv2-Large, a purely vision self-supervised model. This difference introduces a distinct trade-off between alignment and reconstruction. As shown in Section B.2, when VFM-VAE is aligned with DINOv2-Large, its reconstruction quality is slightly weaker compared to alignment with SigLIP2-Large.

To test whether the performance gap stems from the VFM's own strength, we directly aligned VFMs. Following VA-VAE's strong alignment strategy, we trained a VA-VAE with **SigLIP2-Large on 50M images ($\approx$ 40 epochs)**, achieving strong reconstruction and then training the same generative model. Results in Table 5 yield two main conclusions:

- VA-VAE requires substantially longer training to reach competitive alignment and reconstruction, whereas VFM-VAE achieves a balanced trade-off in significantly fewer epochs.
- Even after bridging both the VFM choice and training-scale gap, the VA-VAE variant remains far inferior to VFM-VAE under shorter training, indicating that the performance gain does not simply stem from using a stronger VFM, but rather from **VFM-VAE's design of leveraging VFM features as the starting point**, which provides stronger representational learning ability throughout the generative process.

Table 5: **Differences in reconstruction and generation quality under fair comparison of VFMs and training duration.** CKNNA is computed with own VFM. Generation metrics are reported without CFG.

| Model | VFM | Training Scale | CKNNA↑ | Reconstruction | | Generation | |
|---|---|---|---|---|---|---|---|
| | | | | rFID↓ | IS↑ | gFID↓ | IS↑ |
| VA-VAE | DINOv2-Large | 160M (125 epochs) | 0.202 | **0.30** | **213.6** | 5.14 | 130.2 |
| VA-VAE | SigLIP2-Large | 50M (40 epochs) | 0.099 | 0.84 | 207.4 | 7.83 | 115.1 |
| VFM-VAE | SigLIP2-Large | 44M ($\approx$ 34 epochs) | 0.188 | 0.52 | 208.0 | **3.80** | **152.8** |

## C   HYPERPARAMETER AND MORE IMPLEMENTATION DETAILS.

For VFM-VAE training, the model hyperparameters are listed in Table 7, and the training hyperparameters are listed in Table 8. Our multi-stage training strategy is inspired by VA-VAE (Yao et al., 2025). In the strong alignment stage, we apply relatively large representation regularization losses to quickly establish alignment. In the weak alignment stage, the weight of representation regularization is reduced to maintain alignment while shifting the focus toward reconstruction performance.

We further introduce two fine-tuning components:

- SSIM fine-tuning is motivated by the observation that rapid convergence in reconstruction occasionally causes misalignment across the RGB channels, leading to color noise around edges. To mitigate this, we apply SSIM loss as a refinement.
- PatchGAN (Isola et al., 2017) fine-tuning is added because the original DINO-based discriminator, due to its large patch size, lacks effective supervision on fine details. Introducing a finer-grained PatchGAN improves reconstruction fidelity.

The reconstruction and alignment improvements achieved across the four training stages are summarized in Table 6.

For the generative model, LightningDiT is configured identically to the VA-VAE system. When using REG, we introduce several modifications: the latent size is changed from $32 \times 32 \times 4$ to $16 \times 16 \times 32$, the patch size of SiT-XL is reduced from 2 to 1, the batch size is increased from 256 to 1024, the learning rate is doubled, $\beta_2$ is reduced from 0.999 to 0.95, and QK normalization (Henry et al., 2020) is added in the attention module, all of which are designed to stabilize long-term training. All experiments are trained and evaluated on a single node with $8 \times$ 192GB NVIDIA B200 GPUs.

Table 6: **Reconstruction and alignment performance across four training stages of VFM-VAE.**

| After training period | FID↓ | IS↑ | CKNNA (computed with SigLIP2-Large)↑ |
|---|---|---|---|
| Stage 1: Strong alignment | 1.05 | 198.2 | **0.221** |
| Stage 2: + Weak alignment | 0.60 | 210.3 | 0.188 |
| Stage 3: + SSIM fine-tune | 0.54 | **211.2** | 0.188 |
| Stage 4: + PatchGAN fine-tune | **0.52** | 208.0 | 0.188 |

Table 7: **VFM-VAE Structural Hyperparameters.**

| Category | Name | Value |
|---|---|---|
| VFM Backbone | VFM name | SigLIP2-Large-Patch16-512 |
| Feature | from layers
resolutions
in dims
out dims | [0, 12, -1]
[32, 32, 32]
[1024, 1024, 1024]
[64, 64, 64] |
| Latent | how to compress / decompress
decompress dim
resolution
z dimension | attnproj / attnproj
512
16
32 |
| Concat z | resolutions
mapped dims | [8, 16, 32, 64]
[512, 256, 128, 128] |
| Attention | attn resolutions
attn depths | [8, 16, 32]
[2, 2, 2] |

Table 8: **VFM-VAE Training Hyperparameters.**

| Setting | Strong Alignment | Weak Alignment | SSIM Fine-tuning | PatchGAN Fine-tuning |
|---|---|---|---|---|
| Batch size | | | 512 | |
| Optimizer | | | Adam | |
| Betas | | | (0.0, 0.99) | |
| Learning rate | $1 \times 10^{-4}$ | $1 \times 10^{-4}$ | $5 \times 10^{-4}$ | $5 \times 10^{-4}$ |
| L1 loss weight | 1.0 | 1.0 | 1.0 | - |
| LPIPS loss weight | 10.0 | 10.0 | 2.0 | - |
| DINO discriminator loss weight | 1.0 | 1.0 | 1.0 | 1.0 |
| PatchGAN discrminator loss weight | - | - | - | 1.0 |
| Feature matching loss weight (Isola et al., 2017) | - | - | - | 10.0 |
| SSIM loss weight | - | - | 1.0 | - |
| Multiscale pixel loss weight | 0.1 (to 5M = 0) | - | - | - |
| Representation reglarization loss weight | 5.0 | 1.0 | - | - |
| KL loss weight | $1 \times 10^{-6}$ | $1 \times 10^{-6}$ | - | - |
| Trainable parameters | Entire tokenizer | Entire tokenizer | The decoder | The second half of the decoder |
| Equivariance regularization (Kouzelis et al., 2025) | Yes | Yes | Yes | No |
| Training images | 20M | 20M | 1M | 3M |

# D MORE QUALITATIVE RESULTS

## D.1 RECONSTRUCTION VISUALIZATION

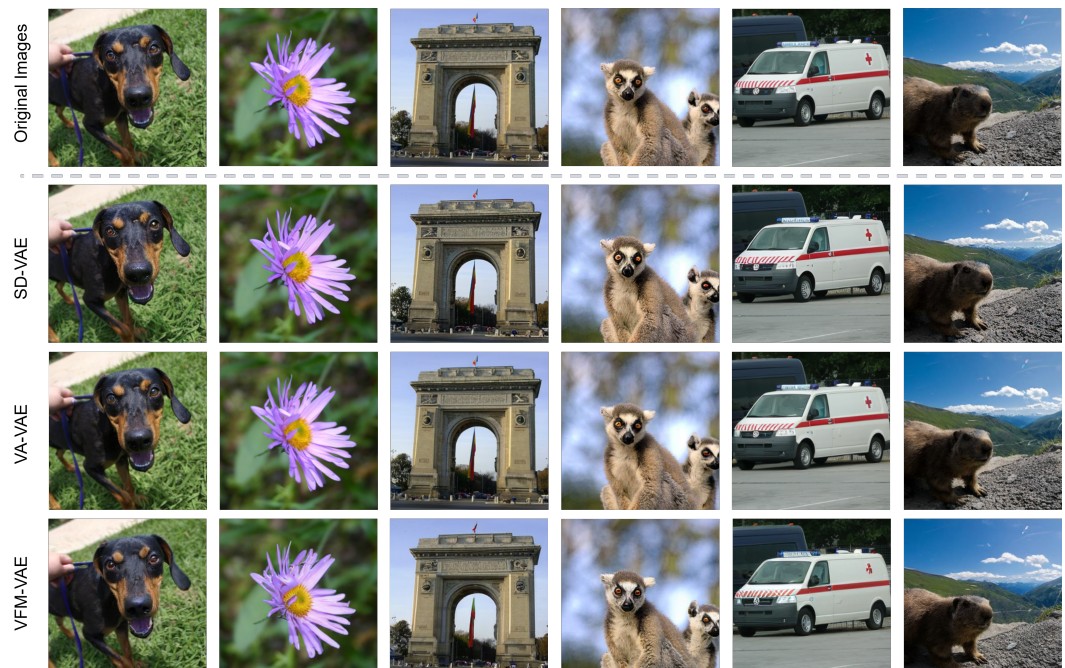

Figure 6: Qualitative comparison of reconstructions from different VAEs.

## D.2 GENERATION VISUALIZATION

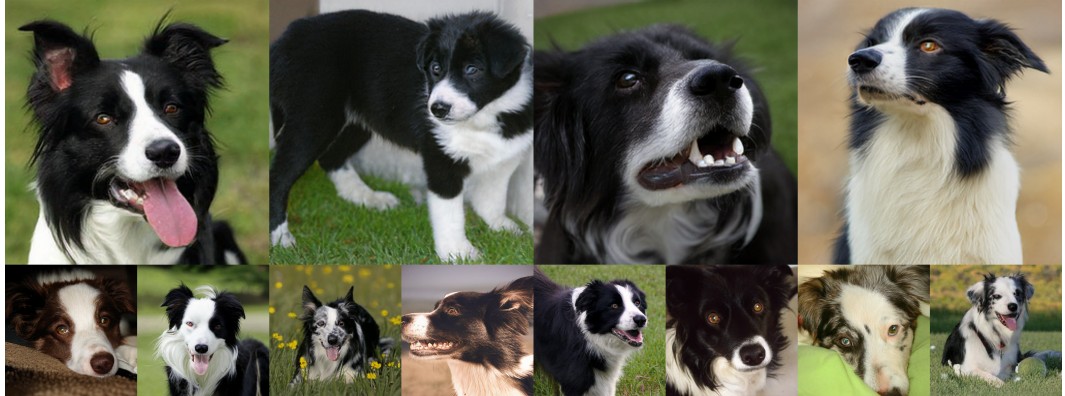

Figure 7: **Visualization of VFM-VAE + REG (640 epochs).** Generation uses CFG with $w = 4.0$; class label is "Border collie" (232).

