# OpenReview forum: "Vision Foundation Models Can Be Good Tokenizers for Latent Diffusion Models"
_ICLR.cc/2026/Conference — ICLR 2026 Conference Withdrawn Submission_

### Official Review · Reviewer_xgJz · 2025-10-18

**Soundness:** 3
**Presentation:** 2
**Contribution:** 2
**Rating:** 4
**Confidence:** 3

**Summary:**

This paper proposes using a pre-trained VFM as the encoder, rather than aligning the VAE latent space with VFMs as in recent works. Specifically, the authors introduce a learnable projection to merge intermediate VFM features into the VAE latent space and design a decoder to hierarchically process the resulting multi-scale features for image reconstruction. The proposed VAE outperforms existing latent-VAE-based generative models in terms of reconstruction, generation, and convergence speed.

**Strengths:**

- Directly utilizing a pre-trained VFM as the encoder is an interesting and novel approach.
- The main empirical results, including those in Table 1, are impressive.

**Weaknesses:**

**Benefits of VFM-VAE over REPA are not clearly articulated**
- Although the proposed method achieves slightly better reconstruction and generation scores (with CFG) than REPA, the improvements are marginal.
- The relationship between representation/generation quality and reconstruction performance is not sufficiently discussed.

**Presentation of the methodology**
- Many components are adopted from previous works, as described in Sections 3.3 and 3.4. While leveraging existing techniques is not a criticism, I strongly recommend that these components be described in detail within this paper for completeness.
- The "reshape" operator should be explained in more detail, as it determines the hierarchical structure of the decoder.
- Although representation regularization loss is a key component, its specific formulation is not provided in the manuscript.

**Motivation for certain techniques is unclear**
- The rationale for employing $L_{VF}$ is unclear. My understanding is that $z$ already contains $f_{final}$ in its representation, albeit mixed with features from other layers via a shallow projection network.
- What is $L_{KL}(z)$? If it refers to the KL regularization term used in standard Gaussian VAEs, the motivation for including this loss should be clarified.

**Ablation study**
- While Table 3 presents some ablation results, the study is not comprehensive. The effects of techniques introduced in Sections 3.3 and 3.4 (e.g., representation regularization loss and multi-resolution reconstruction loss) should be further evaluated.

**Questions:**

- Are there any clear benefits of VFM-VAE over REPA beyond reconstruction and generation performance?
- Why does VFM-VAE achieve strong performance even without CFG? Is there any analysis or discussion on this point?

**Details Of Ethics Concerns:**

Potential ethic concerns are discussed in a section.

---

### Official Review · Reviewer_e6YT · 2025-10-29

**Soundness:** 2
**Presentation:** 2
**Contribution:** 2
**Rating:** 4
**Confidence:** 4

**Summary:**

This paper aims to improve the visual tokenizer of LDMs by directly replacing the VAE with a pretrained encoder (i.e., SigLIP2). The technical contributions lie in the introduction of Multi-Scale Latent Fusion and Progressive Resolution Reconstruction Blocks, which achieve faster convergence and lower gFID on the ImageNet task.

**Strengths:**

1. The paper explores semantic tokenizers, addressing the poor reconstruction quality of semantic tokenizers and improving both generation quality and training efficiency.
2. The proposed modules, including Multi-Scale Latent Fusion and Progressive Resolution Reconstruction Blocks, demonstrate clear empirical improvements in experiments.

**Weaknesses:**

1. The method contains multiple components such as Progressive Reconstruction Blocks, but their motivation and design details are insufficiently discussed. For example, Equations (1) and (2) require more explanation; the rationale for choosing only shallow, middle, and final features (from layers [0, 12, –1]) is unclear and should be justified.

2. Some claims lack explanation or supporting empirical evidence. For instance, line 183: “we recognize that optimal features for reconstruction may not reside solely in the final layer”, and line 346: “Reconstruction ability sets the upper bound for the quality of generation.” These statements are debatable since reconstruction and generation performance are not strongly correlated [1, 2].

3. The model uses SigLIP2-Large-Patch16-512 as a frozen encoder, which was pretrained on billion-scale multimodal datasets [3].
Such large-scale pretraining may already include object-centric distributions similar to ImageNet. It is therefore unclear whether the gFID improvement truly comes from the proposed architecture rather than from the pretrained encoder’s prior exposure to similar data.

4. Although various modules are claimed to improve reconstruction, the reported reconstruction FID increases from 0.3 to 0.5 (tab 2).
This limits the method’s applicability as a general-purpose tokenizer for tasks requiring accurate reconstruction, such as image editing or inpainting.

5. All training and evaluation are conducted on ImageNet, without any out-of-distribution OOD testing. Is it possible that the low gFID arises from overfitting, given that semantic latent distributions are inherently simpler than pixel-level ones [2], rather than from genuine improvements in generalization? Additional experiments are needed to substantiate this claim.

[1] Reconstruction vs. generation: Taming optimization dilemma in latent diffusion models

[2] Masked Autoencoders Are Effective Tokenizers for Diffusion Models

[3] SigLIP 2: Multilingual Vision-Language Encoders with Improved Semantic Understanding, Localization, and Dense Features

**Questions:**

See Weaknesses.

---

### Official Review · Reviewer_Hbk1 · 2025-11-01

**Soundness:** 2
**Presentation:** 3
**Contribution:** 2
**Rating:** 2
**Confidence:** 5

**Summary:**

This paper proposes the Vision Foundation Model Variational Autoencoder (VFM-VAE), a novel visual tokenizer for Latent Diffusion Models (LDMs). The core idea is to directly utilize a frozen pre-trained Vision Foundation Model (VFM) encoder (like SigLIP2-Large) as the VAE's encoder, avoiding the "representation degradation" observed in prior distillation-based methods (e.g., VA-VAE, REPA-E). To enable high-fidelity image reconstruction from the VFM's semantically rich but spatially coarse features, the authors introduce a specialized VFM-VAE decoder featuring Multi-Scale Latent Fusion and Progressive Resolution Reconstruction Blocks. The paper also proposes SE-CKNNA (Semantic-Equivariant CKNNA), a new metric to better evaluate robust semantic alignment. Empirically, the VFM-VAE is shown to accelerate LDM convergence by up to 10x and achieves state-of-the-art FID scores (1.62 without CFG) when combined with an LDM alignment method (REG).

**Strengths:**

* The paper is clearly written and structured. The VFM-VAE architecture, particularly the multi-scale decoder, is well-motivated as a necessary component to overcome the high-level nature of VFM features and achieve pixel-level reconstruction.
* The introduction of SE-CKNNA is a valuable effort to provide a more robust and distribution-aware metric for evaluating latent space quality, which is crucial for diffusion model performance.

**Weaknesses:**

* The claimed superiority of VFM-VAE is highly questionable due to a significant confounding variable: the trainable parameter count (decoder capacity). The ablation study (Table 3) shows the minimal VFM-VAE baseline (43.0M parameters) is nearly unusable (rFID 19.69). Achieving usable performance requires adding Multi-scale Latent Fusion, Modern Blocks, and Encoder Modifications, resulting in a model with 140.6M trainable parameters—more than three times the size of the baseline. It is highly likely that a large portion of the performance gain is simply due to this enormous increase in trainable capacity and architectural complexity
* The new metric, SE-CKNNA, is motivated well but defined vaguely. Section 4.3 mentions applying "a series of semantic-preserving transformation on the evaluated images," including noise, scale-interpolation, and discrete rotations. However, the paper only reports a single SE-CKNNA value in Table 2. This lack of transparency raises two issues: (1) How are the scores from these different transformations (noise, rotation, scale) aggregated into a single, unified SE-CKNNA score? Is it a weighted average, or the minimum score? (2) A new metric should have a stronger theoretical or empirical justification for why these specific transformations are chosen, and why their combination better predicts LDM performance than existing metrics.
* The VFM-VAE is anchored on SigLIP2-Large, while the main baseline VA-VAE used DINOv2-Large. While Appendix B.3 attempts to address this by running a VA-VAE variant with SigLIP2-Large, this comparison remains weak because VA-VAE must learn the alignment from scratch, which naturally requires longer training than VFM-VAE, which starts with the frozen alignment. To truly prove the VFM-VAE design is superior, the authors must show that the architecture is not dependent on the specific, strong SigLIP2 VFM, but generalize well to other VFM features.

**Questions:**

* The increase in trainable parameters from the SD-VAE baseline (43.0M) to the full VFM-VAE (140.6M) is substantial. Please add an ablation where you compare the final VFM-VAE against a VA-VAE or SD-VAE baseline with an equivalently large trainable decoder (i.e., approximately 140M trainable parameters). This is essential to definitively prove that the performance gains are due to the frozen VFM feature injection, and not merely a result of the 3x increase in decoder capacity.
* Given the reliance on SigLIP2's multi-scale features, can the authors demonstrate the robustness of VFM-VAE using an external VFM that is not explicitly designed for dense prediction (e.g., a standard CLIP large model without dense feature training)? This would show whether the VFM-VAE paradigm is generally applicable to "Foundation Models" or only to specific VFMs (like SigLIP2) that already provide well-structured, multi-scale, dense features.

---

### Official Review · Reviewer_wvph · 2025-11-08

**Soundness:** 3
**Presentation:** 3
**Contribution:** 2
**Rating:** 6
**Confidence:** 4

**Summary:**

his paper proposes VFM-VAE, a new approach that directly integrates frozen Vision Foundation Model (VFM) encoders into variational autoencoders used as visual tokenizers for latent diffusion models (LDMs).
Unlike prior distillation-based alignment methods (e.g., VA-VAE, REPA-E) that regressively map VFM features into a trainable encoder, the proposed framework removes this step and uses the VFM itself as the encoder backbone. To compensate for the semantic–fidelity mismatch between foundation encoders and pixel reconstruction, the authors introduce two key decoder components: Multi-Scale Latent Fusion (MSLF) and Progressive Resolution Reconstruction (PRR). Together, these modules enable high-fidelity image reconstruction directly from coarse but semantically rich VFM features.

Additionally, the paper introduces a semantic-equivariant representation diagnostic metric (SE-CKNNA) to study representation dynamics during diffusion training. Experiments on ImageNet-256×256 show strong generative performance (FID = 2.2 without CFG at 80 epochs, 1.62 at 640 epochs), and ablations demonstrate consistent improvement from each architectural component.

**Strengths:**

- Clear motivation and design: The work identifies a concrete limitation—representation degradation from distillation—and replaces it with a direct, conceptually elegant solution: freezing VFMs within the tokenizer.

- Strong empirical results: The proposed VFM-VAE achieves significantly lower reconstruction FIDs and substantially accelerates LDM convergence compared to prior tokenizers such as SD-VAE and VA-VAE.

- Systematic ablation: Table 3 clearly isolates contributions from MSLF, modernized decoder blocks, and encoder feature aggregation, demonstrating each module’s necessity.

- Novel representational analysis: The introduction of SE-CKNNA provides a potentially valuable diagnostic tool for measuring semantic alignment between tokenizer and diffusion model representations.

**Weaknesses:**

- Missing representation evaluation: While SE-CKNNA offers internal diagnostic insight, the paper lacks standard representation quality evaluations, such as linear probing on benchmarks. These would validate whether the frozen VFM representations maintain discriminative structure after reconstruction adaptation.

- Limited generalization evidence: The experiments focus narrowly on ImageNet 256×256; it remains unclear whether VFM-VAE generalizes to higher resolutions or other domains. Particularly, the use of frozen VFM raises concerns about the computation cost when applied to high-resolution image reconstruction. Wall-clock latency comparisons against baselines under variant resolutions are necessary.

- Overemphasis on generation metrics: The evaluation centers on FID and IS, but ignores perceptual metrics (LPIPS, DreamSim), which would more directly connect the model’s goal—semantically faithful reconstruction—to measurable outcomes.

- Ablation study model scales: while the ablation experiment numbers provided in Table 3 seem good. The drastic discrepancy in model size caused by the added components makes the conclusion less convincing.

**Questions:**

- Could the authors include linear probing or related evaluations to quantify whether VFM-VAE preserves the rich semantics of its frozen encoder?

- How sensitive is the generation performance to the choice of backbone (DINOv2 / CLIP / EVA)? Table 4 in the Appendix provides only reconstruction results. However, models that give better rFID do not always deliver better gFID.

- Does the integration of VFM encoders limit resolution scalability or introduce mismatches when training higher-resolution diffusion models (e.g., 512×512)?

---

### Note · Authors · 2025-11-14

I have read and agree with the venue's withdrawal policy on behalf of myself and my co-authors.